



**Atmospheric nanoparticles hygroscopic**
**growth measurement by combined surface**
**plasmon resonance microscope and**
**hygroscopic-tandem differential mobility**
**analyzer**
Zhibo Xie[a,b,c], Jiaoshi Zhang[a,1*], Huaqiao Gui[a,c*], Yang Liu[d], Bo Yang[a], Haosheng Dai[a],
Hang Xiao[b,c], Douguo Zhang[d], Da-Ren Chen[e], Jianguo Liu[a,b,c] .
[a] *Key Laboratory of Environmental Optics and Technology, Anhui Institute of Optics and*
*Fine Mechanics, Chinese Academy of Sciences, Hefei 230031, China*
[b] *Innovation excellence center for urban atmospheric environment of CAS, Institute of*
*Urban Environment, Chinese Academy of Sciences, Xiamen, 361021, China*
[c] *University of Chinese Academy of Sciences, Beijing, 100049, China*
[d] *Institute of Photonics, Department of Optics and Optical Engineering, University of*
*Science and Technology of China, Hefei, Anhui 230026, China.*
[e] *Particle Laboratory, Department of Mechanical and Nuclear Engineering, Virginia*
*Commonwealth University, 401 West Main Street, Ruchmond, VA 23284.*
[1] *now at Center for Aerosol Science and Engineering, Washington University in St. Louis,*
*St. Louis, Missouri, USA.*
* Corresponding author: Jiaoshi Zhang (jszhang@aiofm.ac.cn) and Huaqiao Gui
(hqgui@aiofm.ac.cn)
**ABSTRACT**
The hygroscopic growth of atmospheric aerosols plays an important role in regional



24 radiation, cloud formation and hence climate. Aerosol hygroscopic growth is often

25 characterized by humidified tandem differential mobility analyzers (HTDMA), and Xie et

26 al. (2020) recently demonstrated that hygroscopic growth measurements of single-particle

27 are possible using a surface plasmon resonance microscope-azimuthal rotation illumination

28 (SPRM-ARI). The hygroscopic properties of ambient aerosols are not uniform and often

29 exhibit large RH and size variabilities, due to different chemical compositions and mixing

30 states. To better understand the contribution of different aerosol components and establish

31 the link between the apparent hygroscopic properties of bulk aerosols and single-particle,

32 we conduct combined hygroscopic growth measurements of single-particle by a SPRM-

33 ARI and bulk particles by an HTDMA. The atmospheric nanoparticles were grouped into

34 four subgroups labeled as EC, fly ash, OC and AS+OC based on the energy dispersive

35 spectroscope results (Experimental information: 100nm~200nm, at noon, September 28$^{th}$,

36 2021 and March 22$^{th}$, 2022 in Hefei China). The relationship between the chemical

37 composition of a single nanoparticle in each subgroup and its hygroscopicity was

38 characterized using SPRM-ARI. Then, the HTDMA data were shown to be fitted and

39 reconstructed by the constitutive particle size distributions calculated by the SPRM-ARI

40 measured GFs (growth factor), and the percentage of four subgroups in atmospheric

41 particles could also be found through the fitting. Based on the test results, we found the OC

42 content of AS+OC nanoparticles increased with the increase of particle size, and the OC

43 condensation may play a promoting role in the particle growth process. Lastly, this fitting

44 reconstruction method has a good correlation with the quantitative results of membrane

45 sampling, and can be used for reference to analyze the contribution of particle

46 hygroscopicity and the growth mechanism of nanoparticle.

47

48 **Keywords:** single atmospheric aerosols, hygroscopic growth, surface plasmon,

49 nanoparticles, in situ imaging.

50




## 1. Introduction

The hygroscopicity of aerosol particles plays an important role in the haze and cloud formation, and the climate change in the solar radiation and precipitation (Sloane and Wolff, 1985; Penner et al., 1993). Aerosols can be simply classified as hygroscopic and non-hygroscopic aerosols, which denotes the ability of aerosol particles to form a haze or cloud (Abbatt et al., 2005). In addition, aerosol hygroscopicity can be further complicated due to the mixing or/and heterogeneous reaction of aerosol particles, consequently affecting their abilities in the solar radiation absorption and cloud formation (Shi et al., 2012; Pilinis et al., 1995; Agarwal et al., 2010). The recent research on the atmospheric convective clouds in the Amazon area has shown that nanoparticles can rapidly deliquesce and form the clouds, which further enhances atmospheric convection and promotes the precipitation (Fan et al., 2018). Because of the high concentration, the hygroscopic property of these nanoparticles and its potential contribution to the cloud condensation is believed significant (Tan et al., 2017). It thus becomes very important to understand the relationship between the physicochemical property and the hygroscopic property of these nanoparticles, and the contribution of the above relationship to the overall aerosol hygroscopicity growth process.

The technology for the hygroscopic growth characteristics of aerosols can be classified as: ensemble particle and single-particle. A hygroscopic-tandem differential mobility analyzer (HTDMA) commonly uses to measure the hygroscopic growth of multiple particles in a specific electrical mobility size (Lei et al., 2014; Lei et al., 2018). Although the HTDMA is reliable, the sizes of particles which could be analyzed is typically less than 300 nm (due to the DMA design) and the measured data represent the average of an ensemble particle. For polydisperse particles, the result measured by the ensemble particle technique is the overall average of multiple particle measurements. The technique cannot be used to measure the hygroscopic growth of particles in the large sizes of the size distribution, which has significant contribution to the visibility, weather and climate studies, but cannot be represented by the average (Morris et al., 2016). In contrast, the single-


particle technique offers the qualitative characterization of moisture absorption change due
to the chemical composition and material phase of individual particles (Krieger et al., 2012).
Depending on the particle capture methods, the single-particle hygroscopic growth
imaging technologies can be grouped as the plate-deposition imaging and suspension
imaging (Hiranuma et al., 2008; Peng et al., 2001). The plate-deposition imaging by
traditional 2D imaging methods (*e.g.*, Raman spectroscopy; environmental scanning
electron microscopy, ESEM; surface-enhanced Raman spectroscopy) will be affected by
the imaging angle and orientation(Ebert et al., 2002; Gupta et al., 2015; Craig et al., 2015;
Gen et al., 2017), making it difficult to measure the height change of imaging particles after
the hygroscopic growth. The particle imaging by the current 3D method (i.e., atomic force
microscopy) requires to scan the imaging particle for a long time period, resulting in a long
measurement time and increasing the chance of particle puncture by the scanning probe
(Harmon et al., 2010; Morris et al., 2015). Without the issue encountered in the plate-
deposition imagining (Braun et al., 2001; Lv et al., 2018), the suspension imaging methods
(i.e., electric balance, optical tweezers) are however limited for particles in the sizes larger
than 500 nm, making it very challenging to measure the size change of nanoparticles during
the hygroscopic growth. More, it is difficult to assess the contribution of the hygroscopic
characteristics of single-particle in the atmosphere by the single-particle techniques (Li et
al., 2017; Mikhailov et al, 2015). Although ESEM/environmental transmission electron
microscopy (ETEM) can be used to estimate the concentrations of nanoparticles by
counting them on the ESEM/ETEM images, the process is very time-consuming and only
limited numbers of particles being counted. An efficient and accurate method for
measuring the hygroscopic growth of an ensemble of atmospheric particles is required to
overcome the deficiencies experienced in the current methods.
The surface plasmon resonance microscopy (SPRM) can continuously perform the
imaging measurements of single binding events (Wang et al., 2010; Huang et al.,) and the
light intensity is directly related to the volume of the object (without the photo-bleaching





and fluorophore scintillation) (Young et al., 2018; Halpern et al, 2014). SPRM has been
used widely in the research involving biological targets (including cells, bacteria, viruses,
DNA molecules and proteins) and the local electrochemical reactions and catalytic
reactions of nanomaterials (Syal et al., 2016; Maley et al., 2016; Wang et al., 2012). More,
the influence of the relative humidity on SPRM is very minor, *i.e.*, the imaging is not
affected by the water vapor on the imaging particle surface (Fang et al., 2016). In the
previous study, we used the azimuthal rotation illumination (ARI) to improve the single-
direction SP imaging resolution, which enables a clear imaging of 50 nm PSL particles
(Kuai et al., 2019). The *in situ* SPRM-ARI imaging method was then applied to measure
the hygroscopic growth of 90 nm lab-generated particles (Xie et al., 2020). The above
example demonstrates that the SPRM-ARI can distinguish the size change of particles in
the sizes less than the diffraction limit of the illumination light and accurately characterize
the volume change ratio of nanoparticles after the hygroscopic growth (Kuai et al., 2020).
Therefore, using the SPRM-ARI to perform an accurate hygroscopicity measurement of
atmospheric single nanoparticles and comparing it with the data measured by HTDMA are
expected to provide a rapid method to measure the contribution of hygroscopicity of
individual atmospheric particles to the overall hygroscopic growth of atmospheric particles.
In this work, the *in situ* SPRM-ARI imaging method was used in combination with the
HTDMA technique to characterize the hygroscopic growth of atmospheric nanoparticles
in the sizes of 100, 150 and 200 nm. The chemical compositions of atmospheric
nanoparticles were measured by the scanning electron microscopy (SEM) with an energy
dispersive spectrometer (EDS) and quartz-filter sampling analysis. By recombining the
results measured by the HTDMA and the *in situ* SPRM-ARI, a rapid method was developed
to characterize the hygroscopic contribution of different subgroups atmospheric
nanoparticles.

**2. Materials and methods**



### 2.1 Atmospheric nanoparticle collection and component analysis.


The collection of atmospheric nanoparticles was conducted at the Hefei Institute of
Physical Science (Figure S1), Chinese Academy of Sciences, in Hefei, China (31° 54′ 31″
N, 117° 9′ 36″ E) at the noon of September 28th, 2021 and March 22th, 2022. As shown in
Figure S1, the site is located in the northwest of Hefei City, where both high-temperature
heat sources (thermal power plants) and residential areas are present. The collected
nanoparticles can be considered as representative nanoparticles of inland cities in China.
As shown in Figure 1a, the sampled atmospheric particles were passed through a
diffusion dryer (TSI 3062) to reduce the relative humidity of sampled stream. Particles with
the electrical mobility sizes of 100, 150 and 200 nm were selected by a DMA (TSI 3081)
operated by the DMA platform (TSI 3080). The atmospheric nanoparticles size distribution
in the above two days was shown in Figure S2, and the median particle size was in the
range of 100~150nm, which was consistent with the settled screening particle size.
For atmospheric nanoparticle on September 28th, 2021, classified particles were
collected on the substrate surface in the sample cell. Substrates of two types (in the same
size) were used: one is with the 45-nm-thickness gold coating, which is used for the *in situ*
SPRM-ARI hygroscopic growth measurement, and the other is commercial silicon wafer
used for the SEM measurement. The gold-coated surface (with thickness deviation of ±5%
within a 4 in$^2$ area) was prepared by an e-beam evaporator (K.J. Lesker, Lab 18) on a
standard microscope cover glass (thickness: 0.17 mm) at a vacuum pressure of $<10^{-3}$ mTorr.
The size and element distributions of atmospheric nanoparticles were measured by the
SEM (SU8220, Hitachi, Japan) with an EDS (Aztec, Oxford, UK).
For atmospheric nanoparticle on March 22th, 2022, except the gold coating substrate, the
nanoparticles screened by DMA were collected by quartz filter (Tisch Environmental TE-
20-301QZ). The sampling flow was 1.5L/min, and the collected nanoparticles were used
for organic carbon (OC), elemental carbon (EC) and $SO_4^{2-}$ content measurement. The
content of OC and EC were measured by the traditional thermooptical method (Chow et





al., 2004).In the environment of pure gas He and mixed gas He/O$_2$, the quartz filter
membrane was heated gradiently, and the CO$_2$ produced by catalytic oxidation was
quantitatively analyzed by laser detector (Ding et al., 2014). For the SO$_4^{2-}$ measurement,
the quartz filter membranes with atmospheric nanoparticles were extracted with organic-
free Milli-Q water (Direct-Q3, Millipore)) using an ultrasonic bath for 20 min, and the
content of SO$_4^{2-}$ in the extract was measured by ion chromatography (ICS-3000, Dionex).
The concentration of OC, EC, and SO$_4^{2-}$ reported here were corrected by the blank
membrane, and the concentration was converted into μg/m$^3$.
**2.2 SPRM-ARI hygroscopicity measurement system.**
As shown in Figure 1a, a Nafion dryer, a Nafion humidifier (Perma Pure, USA) and a
proportional-integral-differential (PID) controller were used to control the relative
humidity in the sample cell. The Nafion exchangers enables users to obtain the dry gas (5%
RH) and the wet gas (95%RH). A close loop control with the feedback of measured RH of
the gas (via a RH sensor) was applied by varying the mixing ratio of dry and wet gases to
achieve the set RH. The mixed gas with the desired RH is then directed into the sample
cell to humidify the DMA-classified atmospheric nanoparticles deposited on the Au-coated
surface. The particle deposition and humidification processes were consequently
performed via a two-way switch valve to reduce the interference of impurities during the
growth measurement.
Also shown in Figure 1a is the schematic diagram of the *in situ* SPRM-ARI system,
where the illumination source is a 635 nm parallel laser with the power of 54 mW. Two
orthogonal polarizers are used to eliminate reflected laser signals, allowing the surface
plasmon (SP) signals to be collected as far as possible via a charge-coupled device camera
(Andor, Neo, UK). Using an objective (100×, numerical aperture (NA) of 1.49; Nikon,
Japan) and a pair of scanning galvanometers, the laser beams could be focused on any
position in the back focal plane (BFP). In this configuration (as shown in Figure 1b), the
laser beam can rotate around the Au film at a specific angle ($\theta$), which is called as the





azimuthal rotational illumination. Figure S3 shows the reflection BFP image of the 45 nm
Au film. The presence of the symmetrical dark arc on the image verified the existence of
the p-polarized SPs. The SP signals of atmospheric nanoparticles on the Au film were
recorded as the cell RH was increased. Under the ARI mode, split circular spots are formed
if the particle size is less than the diffraction limit. By combining with the information of
the DMA classification and SEM measurement, the initial sizes of selected atmospheric
nanoparticles can be clearly determined (Xie et al., 2020). The statistics of the gray
intensity (GI) on the SP images is applied for the SP image processing (Huang et al., 2007).
It is because the GI is positively correlated with the volumes of imaging particles, and the
GF (growth factor) of imaging atmospheric nanoparticles can be obtained by the cube root
of the GI.

For the reference to the Köhler theory (Petters et al., 2007; Fan et al., 2020), the

hygroscopicity parameter $\kappa$ can be calculated using the GF measured by the SPRM-ARI
system.
$$\kappa = \left( \frac{\exp\left(\frac{A}{D_d Gf}\right)}{RH} - 1 \right)(Gf^3 - 1) \tag{1}$$
$$A = \frac{4\sigma_{s/a} M_w}{RT\rho_w} \tag{2}$$

where $Gf$ is the growth factor measured using the SPRM-ARI system, $D_d$ is the dry

diameter of the atmospheric particles, $RH$ is the relative humidity in the sample cell (RH:
84%) and $\sigma s/a$ is the surface tension of the solution/air interface ($s/a$=0.0728 $\mathrm{Nm^{-2}}$). $M_w$ is
the molecular weight of water, $R$ is the universal gas constant, $T$ is the environmental
temperature (298 K) and $\rho_w$ is the water density. The deliquescence droplet was selected
as the default well-mixed solution. Note that the Zdanovskii-Stokes-Robinson (ZSR)
model was not used to calculate the $k$ value because of the complex chemical composition
of atmospheric particles.
**2.3 HTDMA**

The HTDMA was used to measure the hygroscopic growth of the above two days

atmospheric nanoparticles in a narrow electrical mobility size distribution as those



measured using SPRM-ARI. Figure S4 shows the HTDMA including a long DMA (TSI
3081), a humidification chamber and a scanning mobility particle sizer (TSI DMA 3081
and WCPC 3788) system. After passing through the diffusion dryer and particle charge
neutralizer, atmospheric particles were introduced into the DMA1 to select atmospheric
particles in the electrical mobility sizes of 100 nm, 150 nm and 200 nm. Atmospheric
particles in the selected sizes were humidified in a Nafion pipe (at the flow rate of 0.3
L/min) with a sheath gas who's RH is controlled in the same manner as that in the SPRM-
ARI humidification system. The same RH-controlled gas was also used for the sheath flow
for the DMA2 operation. The sheath gas of the DMA2 was circulated via an air pump with
an air filter (Parker 9922-11-AQ).

**3. Results and discussion**
**3.1 Combined SPRM-HTDMA measurements of 100 nm ambient aerosols**
Figure 2 shows the SEM images of typical 100nm particles and their EDS mapping results
on September 28$^{th}$, 2021. According to the morphology and EDS mapping, atmospheric
particles were grouped into four subgroups labeled as: EC (Figure 2a), OC (Figure 2b), fly
ash (Figure 2c) and AS+OC (ammonium sulfate) (Figure 2d). The EDS of the 100 nm
atmospheric nanoparticles (given in Figure 2a) showed the dominant C element signal and
no obvious presence of the O element signal. It can be concluded that the 100 nm
atmospheric particulate shown in Figure 2a shall be in the EC group, which are mainly
produced by the incomplete combustion of fossil fuels (Jacobson et al., 2000). As for
nanoparticles in the OC group (in Figure 2b), an obvious O element signal was present in
addition to the obvious C element signal. The sources of OC particles are very diverse. The
OC particles could be either from the direct emission of pollution sources or the secondary
formation of atmospheric volatile organic compounds (VOCs) (Zhang et al., 2017). In
Figure 2c, the EDS mapping was mainly the Fe and O element signals. Note that the Si
element signal cannot be used to identify the particles because of the Si substrate. The





particle morphology in Figure 2c shows a cluster of spherical particles, and the Fe and O
element signals are evenly distributed over the whole SEM-ed particle. The particle shown
in Figure 2c is thus grouped as fly ash. The source of fly ash particles is likely from the
high temperature combustion (Bondy et al., 2018). Atmospheric nanoparticles, shown in
Figure 2d, are ones containing OA and AS particles, which are widely distributed in rural
and urban areas. The EDS of the OA+AS particles shows obvious S element signals in
addition to the C and O element signals. The unique characteristics of the particle
morphology (shown in Figure 2d) is the brighter color, which is due to the presence of
sulfate (because of the high conductivity of AS particles). The image also evidenced that
the mixing of OC and AS in the particle is not uniform in 100nm atmospheric nanoparticles.

Figure 3 shows the measurement of the hygroscopic growth of 100 nm atmospheric

particles by *in situ* SPRM-ARI. The SPRM grayscale images are provided at the RH levels
of 30%, 60%, 80% and 90%. By collecting the GI of the SPRM images under various RH
conditions, the cube root ratio of the GI can represent the GF for the particle moisture
absorption growth. It is found that, during the hygroscopic growth process of 100 nm
atmospheric nanoparticles, the *in situ* SPRM-ARI results were split circular spots, similar
to those obtained for the previous *in situ* SPRM-ARI obtained for the hygroscopic growth
of nanoparticles in a pure composition (Xie et al., 2020; Kuai et al., 2020). The above
observation also indicates that the atmospheric nanoparticles involved in this work were
less than the diffraction limit. The size of the measured particles was determined to be 100
nm by SEM.

As the RH increased, the *in situ* SPRM-ARI results can be grouped into three growth

types depending on the GI variation. In the first growth type, the GI did not obviously
change, and the speckled spots on the SPRM images basically remained their shapes at low
RH, indicating that the GF of this 100 nm atmospheric particle did not change as the RH
increased. According to the previous experience, this series of *in situ* SPRM-ARI results
should be for particles either in the EC or fly ash group. In the second growth type, the GI



of split circular spots was gradually changed, *i.e.*, the GI of the spots slowly increased while
the circular spots remained segmented, indicating the size of the atmospheric particle
during the hygroscopic growth remained less than the diffraction limit. Compared to that
obtained for particles in the first growth type, the SPRM-ARI results of particles in this
type shows the brightness of the spots had been increased. In other words, the GF of
atmospheric particles in this group was small, and their sizes were slowly increased during
the hygroscopic growth. The GF of this nanoparticle was 1.25 when the RH reached 90%.
Based on the previous result[42], this atmospheric particle should be in the OC group of
atmospheric particles. The last growth type for the SPRM-ARI result is for the cases where
the shape of split circular spots had obviously changed during the hygroscopic growth, i.e.,
the intensity of the spots had become bright and the split segments of the spots had merged,
indicating that the particle size has obviously changed. No obvious phase transition of the
particle was however observed during the hygroscopic growth of particles in this last
growth types, *i.e.*, no obvious deliquescence relative humidity (DRH) detected. The GF
was 1.4 when the RH reached 90%. Particles in the last growth type should be in the
AS+OC group of atmospheric particles and the deliquescence of the AS has obviously been
affected by the presence of liquid OC.
Subsequently, the 100nm bulk hygroscopic growth was analyzed synchronously by
HTDMA at 84% RH (i.e., after the hygroscopic growth Fig4). The particle size
distributions of 100nm EC/fly ash, OC, and AS+OC particles after the hygroscopic growth,
calculated using the measured GF, were also shown in the same figure. The HTDMA-
measured size distribution was also fitted by the linear combination of the size distributions
of EC/Fly ash, OC and AS+OC particles (after the hygroscopic growth), and the result of
the fitted size distribution is also given. By the fitting reconstruction, the percentages of
EC/Fly ash, OC and AS+OC particles in the HTDMA-measured size distribution can be
obtained. It is found that 100 nm atmospheric particles primarily consisted of OC and
AS+OC particles, and the ratios of OC and AS+OC particle areas were 45.9% and 47.1%,





respectively.

In this part of the study, we classified the 100 nm atmospheric particles into four groups

using the SEM images and EDS spectrum on September 28$^{th}$, 2021. The GFs of these
nanoparticles had then been measured by *in situ* SPRM-ARI and HTDMA. The
hygroscopic growth of these 100nm particles in different groups can be observed at the
ambient pressure by the SEM and SPRM-ARI techniques, and the proportion of different
groups can be obtained by HTDMA peak fitting reconstruction.
**3.2 Size-dependence of SPRM-HTDMA derived chemical composition.**
Figure 5 and Figure 6 shows the SEM and SPRM-ARI results of atmospheric particles in
the 150 nm and 200 nm size on September 28$^{th}$, 2021, and their associated EDS mapping
results are given in Figure S5-S6. Like the 100 nm atmospheric particles, 150nm and
200nm atmospheric particles can also be classified into four groups according to their
SEM+EDS mapping results. Compared with the results for the 100 nm atmospheric
nanoparticles, no obvious difference in each group was found. The only observable
changes were found in the SEM images of particles in the fly ash and AS+OC groups. The
clustering status of particles in the fly ash group and the asymmetric shell structure of
particles in the AS+OC group becomes very obvious. Especially for 200 nm particles, the
AS was found only at the center of these particles (while the OC was widely distributed).

Figure 5b and 6b shows the SPRM-ARI results for 150nm and 200nm atmospheric

particles. The SPRM imaging results of these particles (shown in Figure S6 and S8) can
also be grouped into three growth types: no obvious gray signal change for EC/fly ash
nanoparticles; the gray signal enhancement but no obvious shape change in the gray
circular spots for OC nanoparticles; and gray signal enhancements and clear shape fusion
for AS+OC nanoparticles. When compared with the SPRM-ARI results of 100 nm particles,
no observable change in those EC/fly ash and OC particles. However, for the AS+OC
particles, their hygroscopic GF decreased as the particle size increased, particularly in the
high RH range (>80%). Especially for 200nm AS+OC particles, the GF at the 90% RH





decreased from 1.58 to 1.46. Combined with SEM images and EDS mapping results, he
reason for the decrease of GF may be the gradual increase of organic content in AS+OC
particles.
To clearly show the growth trend of AS+OC particles, the kappa ($\kappa$) values for the
particle sizes of 100-200 nm were calculated at 84% RH (given in Figure 7). At the 84%
RH, AS+OC particles would have been completely deliquesced under the assumption of
homogeneous liquids. The $\kappa$ value of pure ammonium sulfate (AS) particles in the same
size range was also given. It is found that the change of the $\kappa$ value for OC and EC/fly
ash particles is negligible in the calculated size range. For AS+OC particles, the $\kappa$ value
decreases with the increase of particle size, which indicates that the proportion of organic
components (OCs) contained in these particles gradually increases. Thus, the above
observation of the $\kappa$ value for AS+OC particles further confirms that, in the studied size
range of the aggregates (100-200 nm), the OC was grown on the sulfate core by the
adsorption and condensation.
Subsequently, the HTDMA was also used to analyze bulk hygroscopic growth of 150nm
and 200nm atmospheric particles at 84% RH, and the hygroscopic particle size spectrum
was fitted and reconstructed (Figure 8). It can be see that, as the dry atmospheric particle
size increased, the peak separation for OC and AS+OC particles after the hygroscopic
growth have changed obviously. The percentage of OC particles in selected atmospheric
particles gradually increased and reached 78.2% for 200 nm atmospheric particles. And,
the percentage of AS+OC particles in selected atmospheric particles gradually decreased
and , *i.e.*, most of AS+OC nanoparticles occurred at the border between the Aitken (10 nm-
100 nm) and the condensation modes (100 nm-300 nm). The increase of the OC content in
AS+OC particles observed in both the SEM and SPRM measurements also concluded in
the HTDMA analysis, *i.e.*, the AS+OC hygroscopic particle peak moved to small particle
size as the increase of the dry particle size, which also indicated that OC an important role
in the AS+OC nanoparticles growth.



### 3.3 Comparisons of OC, EC and $SO_4^-$

In order to determine the feasibility of combining SPRM atmospheric single nanoparticle test results with HTDMA to evaluate OC, EC and AS+OC contents, the atmospheric nanoparticle hygroscopic growth characteristics on March 22, 2022 have been tested again, and the OC, EC and $SO_4^-$ content of different particle sizes were quantitatively analyzed. Figure S9 was the SPRM test results of 100nm, 150nm and 200nm single-particle hygroscopic growth. It can be seen that the atmospheric particulates can also be divided into three categories according to the hygroscopic capacity: EC/flyash without hygroscopic growth, OC with weak hygroscopic growth, and AS+OC with strong hygroscopic growth, and with the increase of particle size, the hygroscopic capacity of AS+OC nanoparticles decreased gradually. Comparing the hygroscopic growth results of atmospheric nanoparticles in two days, we found the GF of AS+OC nanoparticles on March 22, 2022 was higher, which may be due to the more $SO_4^-$content of the mixed particles in this experiment. Then, based on the SPRM measurement results, the HTDMA hygroscopic particle size spectrum in the same period could also be divided into three normal distribution peaks (figure S10) and the GF results of SPRM are consistent with the peak position of HTDMA at RH84%. Compared with fitting results on September 22, 2021, the number concentration of atmospheric nanoparticles on March 22, 2022 was lower, and the peak distribution of the three categories were relatively independent. According to the peak area ratio, we found, with the increase of particle size, the proportion of EC was basically about 14%, the proportion of OC gradually increased from 20% to 29%, and the content of AS+OC gradually decreases from 65% to 56.4%, which indicated that the condensation of OC could play an adhesive role in the growth of AS+OC nanoparticles. Subsequently, by measuring the EC, OC and $SO_4^-$ contents of nanoparticles collected by the quartz film at the same period (Figure 9), we found, with the increase of particle size, the OC content increased from 33% to 41.8%, and the $SO_4^-$ content decreased from 56% to 46.96%. In contrast, the OC content obtained by HTDMA peak splitting method was slightly less than

https://doi.org/10.5194/acp-2022-666Preprint. Discussion started: 18 October 2022
[Figure]

that of the quartz film sampling, which may be due to the OC content obtained by quartz
film have included OC in AS+OC category and OC without hygroscopic capacity in
EC/flyash category (such as dicarboxylic acid nanoparticles). Comparing the HTDMA
peak fitting results and the quartz film sampling results, the analysis method based on the
classification of particle hygroscopic characteristics is reliable, which may be helpful to
analyze the hygroscopic contribution and growth mechanism of different types
atmospheric nanoparticles.

**4. Conclusions**

The measurement of the hygroscopic growth of single nanoparticles is important for the
analysis of the chemical composition and hygroscopic characteristics of atmospheric
particles, and for the scientific studies involving atmospheric particles. The SPRM imaging
technology offers its advantages in the hygroscopic growth measurement of single
nanoparticles over the existing methods (e.g., SEM and AFM), particularly for irregular
shaped particles. It is because the gray signal of the SPRM-ARI imaging can be positively
correlated with the volume of the imaged particles.
In this work, atmospheric nanoparticles in the electrical mobility sizes of 100, 150 and
200 nm were classified by a DMA and collected for both single-particle and bulk
hygroscopic growth measurements. For single-particle measurements, based on the SEM-
EDS data, atmospheric particles in a selected size could be grouped into four enabled as
EC, fly ash, OC and AS+OC groups. The hygroscopic growth factor of particles in each
above group were measured by the SPRM-ARI, and the hygroscopic growth of
atmospheric particles due to the very minor change in the chemical composition (e.g., AS+
OC particles) could be detected by the above measurements.
For bulk hygroscopic growth measurement, this work further demonstrates that the
HTDMA-measured size distribution of atmospheric particles at a high RH (i.e., after the
hygroscopic growth) could be reconstructed by the linear combination of the calculated



size distributions of constitutive particles at a high RH. Besides, the HTDMA peak fitting experiment on March 22, 2022 shows that there was a good correlation between the method of HTDMA peak fitting reconstruction and the quartz film sampling. And, the experimental results of atmospheric nanoparticles hygroscopic growth for two days show that the OC condensation plays an important role in the growth of AS+OC nanoparticles. This reconstruction method has potential significance in predicting the contribution of atmospheric particulate hygroscopicity and particle growth mechanism.

**Author contributions**

**Zhibo Xie:** Methodology, Validation, Visualization, Writing – original draft, Writing – review & editing. **Huaqiao Gui:** Conceptualization, Resources. **Jiaoshi Zhang:** Funding acquisition, Methodology. **Yang Liu:** Methodology. **Bo Yang:** Investigation. **Haosheng Dai:** Data curation. **Hang Xiao:** Visualization. **Douguo Zhang:** Methodology, Writing – original draft, Investigation. **Da-Ren Chen:** Conceptualization, Validation. **Jianguo Liu:** Conceptualization, Resources, Funding acquisition.

**Competing Interest**

The authors declare no competing financial interest.

**Financial support**

This work was supported by the National Natural Science Foundation of China (41905028, 91544218), the National Key R&D Program of China (2017YFC0209504), the Science and Technological Fund of Anhui Province (1908085MD114, 2108085MD139), and the HFIPS Director's Fund (Nos. YZJJ2022QN04, BJPY2021A04).

**Supplement**

Figures S1−S10 are described in the text (a PDF is available).



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

**Figures Caption**





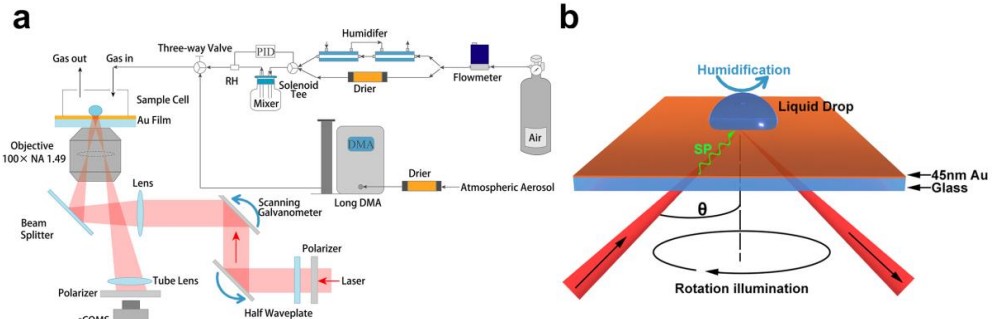


Figure 1. Schematic diagram of the *in situ* SPRM-ARI single nanoparticle moisture
absorption system: (a) for the complete system setup, and (b) for the gold-coated glass

substrate used for *in situ* SPRM-ARI.

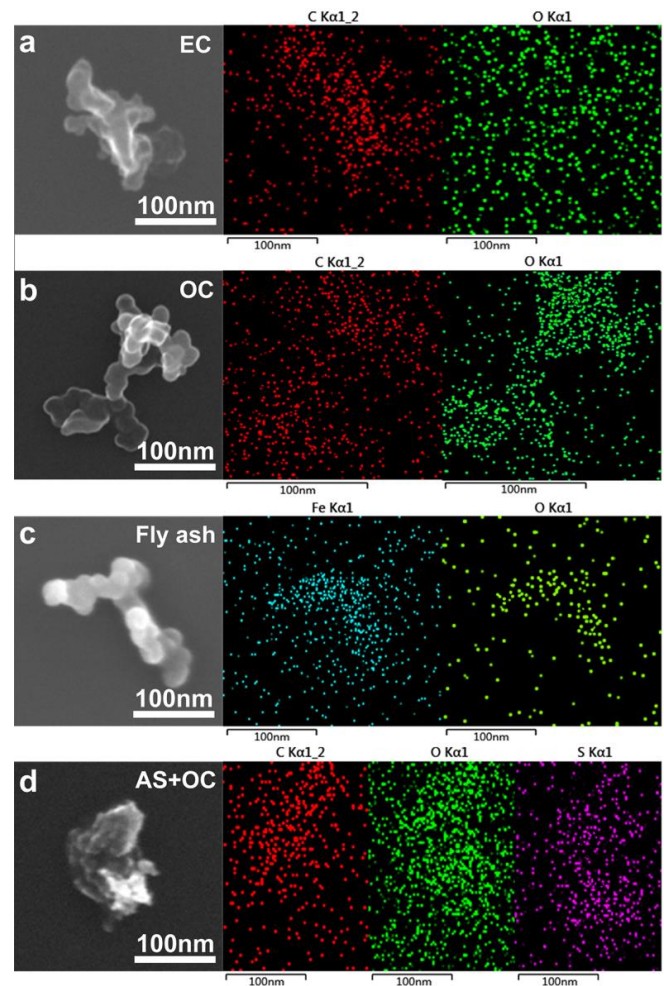


Figure 2. SEM and EDS mapping of typical 100 nm atmospheric particles collected in
this study.



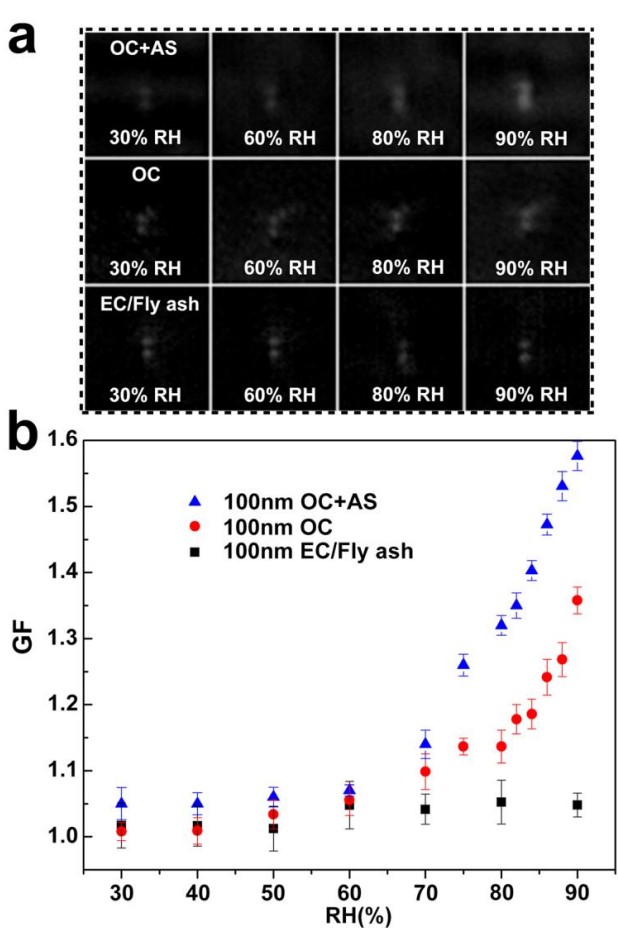

Figure 3. (a) SPRM-ARI images and (b) hygroscopic growth factors of 100 nm atmospheric particles.






Figure 4. HTDMA and peak fitting reconstruction for 100 nm atmospheric particles at
84% RH on September 28th, 2021.

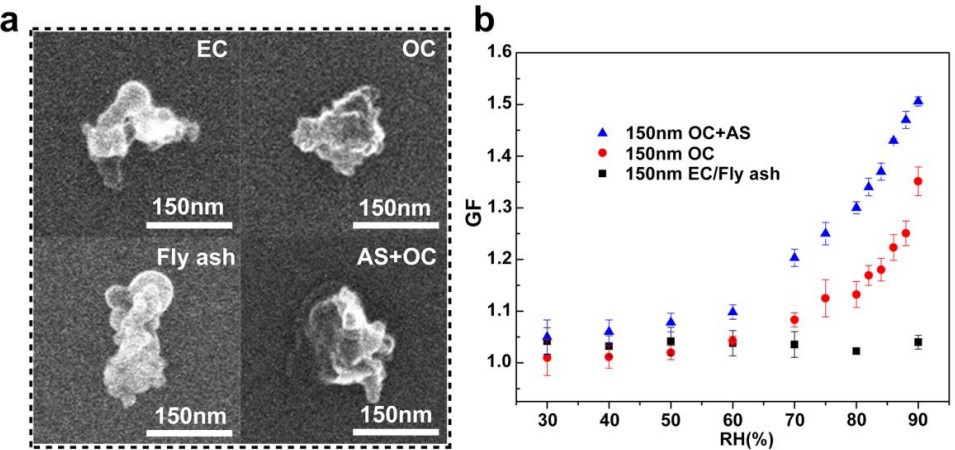


Figure 5. (a) SEM images and (b) hygroscopic growth factors of atmospheric particles in
the 150 nm size.



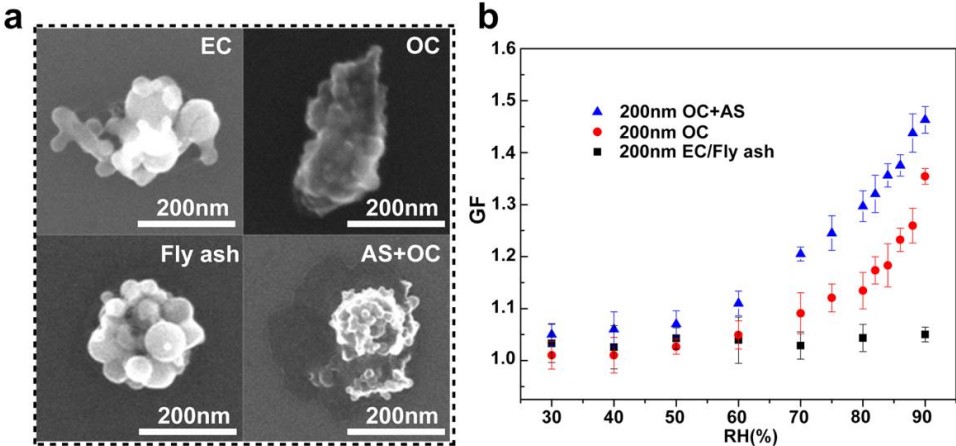


Figure 6. (a) SEM images and (b) hygroscopic growth factors of 200 nm atmospheric

particles.

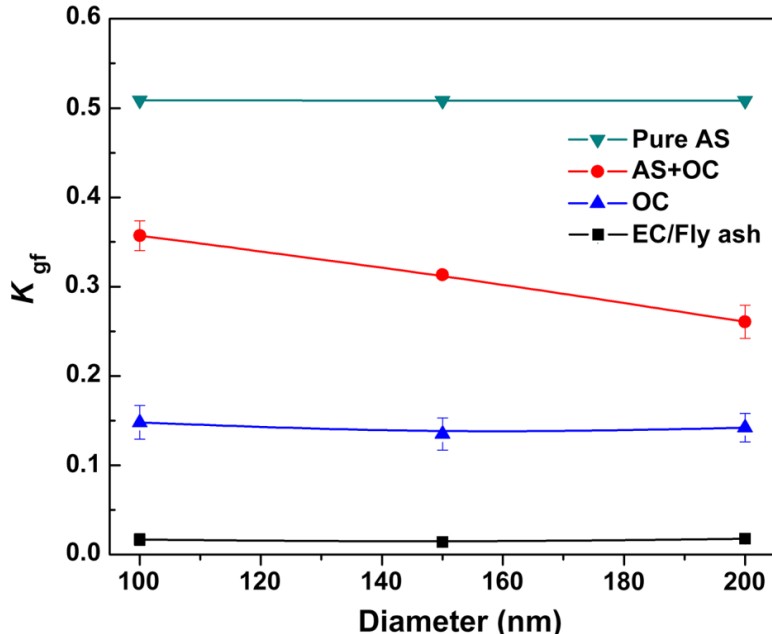


Figure 7. $\kappa$ results for the 100 nm, 150 nm and 200 nm atmospheric particles at RH on
September 28th, 2021.





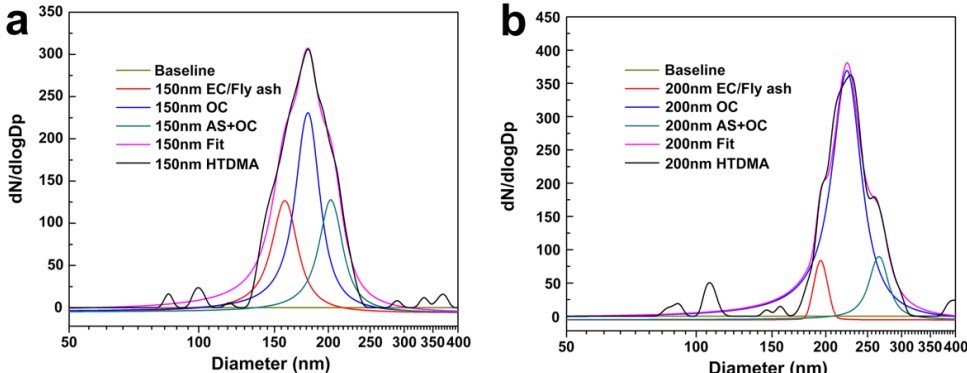


Figure 8. HTDMA and peak fitting reconstruction for (a) 150 nm and (b) 200 nm

atmospheric particles at 84% RH on September 28th, 2021.

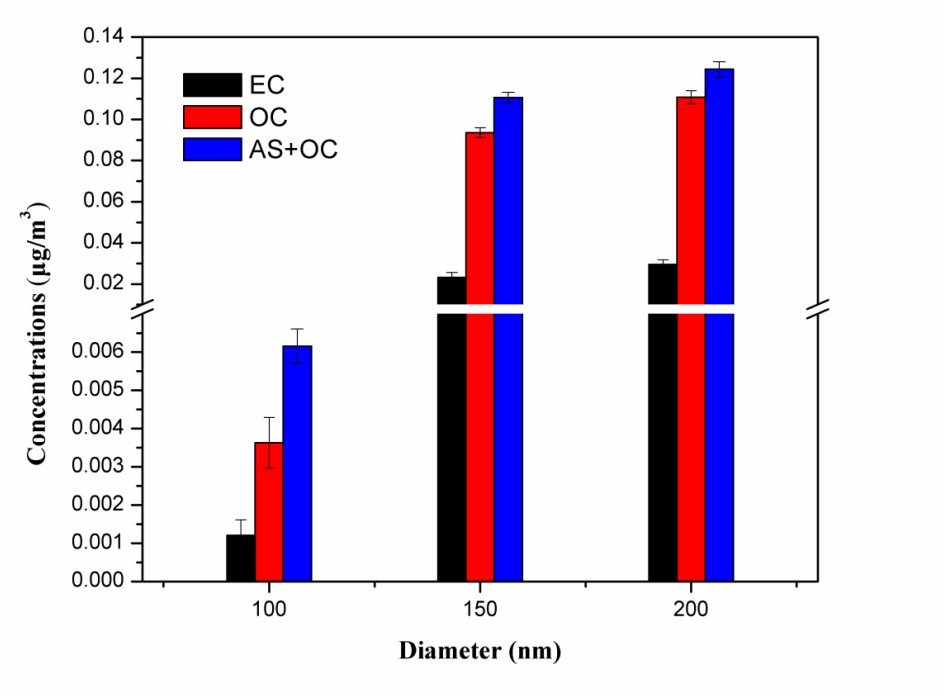


Figure 9. Quantitative results of atmospheric nanoparticles subgroups collected by quartz

filter membrane on March 22th, 2022.
