# Peer review of "Atmospheric nanoparticles hygroscopic growth measurement by combined surface plasmon resonance microscope and hygroscopic-tandem differential mobility analyzer"

_Atmospheric Chemistry and Physics, 2022_

## Author Comment (AC1)

**Manuscript No.**: ACP-2022-666

**Title**: Atmospheric nanoparticles hygroscopic growth measurement by combined surface plasmon resonance microscope and hygroscopic-tandem differential mobility analyzer

We thank the anonymous referees for their valuable and constructive comments/suggestions on our manuscript. We have revised the manuscript accordingly and please find our point-to-point responses below.

**Comments by Anonymous Referee #1:**

*General Comments:*

*The manuscript "Atmospheric nanoparticles hygroscopic growth measurement by combined surface plasmon resonance microscope and hygroscopic-tandem differential mobility analyze" by Xie et al. shows the new coupling of SEM-EDX/HTDMA and SPRM (also combined with chemical measurements of EC, OC and sulfate components) for the investigation of hygroscopic growth of real ambient 100nm, 150nm and 200 nm particles.*

*Such measurements are important and fits well into the scope of ACP, even when the methodological/instrumental part takes a large part in this manuscript and should actually be even longer in order to be able to follow all the details.*

*Partly the explanations are not fully comprehensible, also because some terms are not explained and some occurring abbreviations are explained only in the later part of the manuscript.*

**Response:** We thank the reviewer for the constructive comments and suggestions. Point-to-point responses to comments and questions are detailed below. Following the reviewer's suggestions, we organized the manuscript in the clearer way and clarified the significance of combined SPRM and HTDMA measurements for particle hygroscopic growth studies. The new results and discussions are now included in the revised manuscript. We checked all the abbreviations in the manuscript and ensured that their full names are introduced where they first appear.

*Major Comments:*

*This is most critical for the main SPRM-ARI method. For the essential details, reference is made to other papers and the only visible result of the method is a very fuzzy Figure 3a. No information is given in the manuscript about the number of particles studied, so statistical significance is difficult to estimate.*

**Response:** We thank the reviewer for this suggestion. We adjusted the contrast of the plots in Fig. 3 and a clearer version is provided in the revised manuscript. In this way, the variation in gray intensity is now easier to distinguish. We have recently demonstrated that hygroscopic growth measurements of single-particle are possible using the SPRM-ARI, specifically lab-generated particle standards (Xie et al., 2020). In this study, we applied the SPRM measurement to atmospheric particle samples, and classified the individual particles into different categories according to their distinct hygroscopic properties.

*Most of the fundamental data and conclusions are obtained from SEM and HDTMA measurements, so the paper as presented is more of a SEM/HDTMA as a SPRM-ARI/HDTMA coupling.*

**Response:** In this study, we aim to conduct combined hygroscopic growth measurements using a SPRM-ARI and an HTDMA and establish a link between the apparent hygroscopic properties of single particles and bulk aerosols. In order to do that, we first identified individual particles with distinct hygroscopic growth behaviors from the SPRM single-particle probing and classified those particles into different categories including non-hygroscopic (NH), less-hygroscopic (LH), and more-hygroscopic (MH). Next, the mean growth factor (GF) of the three categories can be utilized to reproduce the GF distribution obtained from the HTDMA measurement, such that the number fractions of the three categories can be retrieved. To achieve a hygroscopicity closure, we identified the chemical compositions of individual particles using SEM/ESD analysis, and the results likely agree with the apparent hygroscopic properties of individual particles from SPRM measurement.

*The presented main results of the manuscript are:*

*Establish a link between hygroscopic properties of bulk aerosol and single particles respectively establishing a link between single particle composition and its hygroscopicity.*

*The OC content of larger mixed AS/OC particles (100 nm vs. 200 nm diameter) increases.*

*The used fitting reconstruction method has a good correlation with quantitative determined OC, EC and sulphate concentrations.*

*To 1) I am not sure if this link is reached in the manuscript by SPRM-ARI. Due to the missing statistics and the fuzzy Figure 3a, the mentioned advantage of SPRM-ARI over other methods mentioned in the manuscript (ESEM/ETEM) is not clear enough (see following discussion).*

**Response:** Following the reviewer's suggestion, we adjusted the contrast of the plots in Fig. 3 and a clearer version is provided in the revised manuscript. In this way, the variation in gray intensity is now easier to distinguish. We have recently demonstrated that hygroscopic growth measurements of single-particle are possible using the SPRM-ARI, specifically lab-generated particle standards (Xie et al., 2020). Specifically, the statistics of the gray intensity (GI) on the SPRM images is

directly related to the volume of imaging particles, the size of the examined particle can be obtained by taking the cube root of the GI. In this way, we examine the particle size change under different RH levels, and the hygroscopic GF can be derived accordingly. We clarified this in the revised manuscript.

Compared with other optical microscopy approaches, SPRM-ARI can quickly and nondestructive measure the change of particle physical and chemical properties under normal pressure and temperature conditions, while still maintains the high sensitivity of optical microscopy. On the other hand, for ESEM/ETEM, the viewing direction is typically perpendicular to the substrate plane, making it non-easy to measure the height of imaging particles accurately. Besides, the electron beam may damage the particle, especially for multicomponent particles. This has been systematically investigated in our previous study of hygroscopic growth of lab-generated particle standards. (Xie et al., 2020; Kuai et al., 2019).

*To 2) This observation seems to be correct for the 2 collections carried out at the specific sample location. To derive a general pattern from this is not permissible.*

**Response:** Thank for the reviewer's comment. We acknowledge that the 2 collections cannot represent the general case in the atmosphere. We clarified this in the manuscript as:

"It is clear that the increase of OC compounds is reflected in both the coupled SPRM-HTDMA measurement and the chemical analysis results, which suggests that the condensation of organic

compounds plays an important role in the hygroscopic growth behavior, particularly for the two experiments we conducted."

*To c) I cannot share this statement from the data shown (also due to the points still to follow). I agree that there is no contradiction but there is not enough data given to make a correlation visible.*

**Response:** We combine chemical analysis from collected aerosol samples on March 22th, 2022 to reinforce the hypothesis that the size-dependent hygroscopic properties can be explained by the variation in OC compounds. We only focus on the size-dependence of OC, EC, and $SO_4^{2-}$ compounds, and we sum up the OC, EC and $SO_4^{2-}$ concentrations and normalized to 1. Since our preliminary analysis suggests an increase of OC compounds with increasing particle sizes from 100 to 200 nm, we claimed that the increase of OC compounds is reflected in both the coupled SPRM-HTDMA measurement and the chemical analysis results.

We did not perform direct correlation analysis; therefore, it was not rigorous to say that they have good correlations, and we modified the corresponding descriptions as the reviewer suggested.

*Further points:*

*Definition of subgroups: For the given particle diameters (100-200 nm) of an urban aerosol typically mixtures of secondary material (organic, nitrates and sulfates) and soot (which is a mixture of OC and EC) dominates. Often many of these components are internally mixed and the hygroscopic behavior of this mixture is given by the HTDMA curve in Figure 4.*

*Following the secondary electron images given in Figures 2 (figure2 legend is erroneous), 5 and 6 all shown particles (except the fly ashes) seems to be dominantly soot, respectively mixtures of soot and secondary material. As soot is a mixture of OC and EC components this does not necessarily contradict the given EC, OC subgroup definition. As a simplification, the approach of classifying all carbon-rich particles with low oxygen content as soot (dominant EC – will show no strong water uptake) and those with very high oxygen content as OC (low or no soot content) may be permissible. But the simplification of all secondary material as ammonium sulfate does not seem permissible to*

*me. Maybe it should be called ambient secondary material. The shown EDX mappings in Figure 2 are not helpful for the proof of ammonium sulfate as the shown count rates are too low and nitrate cannot be detected in EDX as a nitrogen peak may originate from ammonium or nitrate.*

**Response:** We thank the reviewer for this suggestion.

According to the suggestions of the reviewer, we have adjusted the SEM classification of atmospheric particles in this manuscript. For EC component, it is modified to soot (mainly EC), and AS+OC is modified to secondary aerosol (mainly OC and $SO_4^{2-}$).

*The division of the HDTMA curve into 4 sub curves based on the 4 self-defined subgroups seems uncertain to me because of the problems mentioned above. Also, it does not seem clear to me to what extent the SPRM-ARI data played a role here. The significance of these measurement must be shown and worked out more clearly or the statements must be adjusted accordingly.*

**Response:** We first identified individual particles with distinct hygroscopic growth behaviors from the SPRM single-particle probing and classified those particles into different categories including non-hygroscopic (NH), less-hygroscopic (LH), and more-hygroscopic (MH). Next, the mean growth factor (GF) of the three categories can be utilized to reproduce the GF distribution obtained from the HTDMA measurement, such that the number fractions of the three categories can be retrieved.

HTDMA measured aerosol hygroscopic GF distribution is normally classified into two modes, with the first mode being recognized as "less hygroscopic" or "hydrophobic", and the second mode being recognized as "less hygroscopic" or "more hygroscopic", depending on the mode GFs. However, the mode separation is not always ideal. Sometimes, one of the two modes could be very flat, meaning that it may include particles with quite different hygroscopicities. In this case, by investigating at the individual particles, we would get extra information about the factors affecting the apparent hygroscopic growth behaviors, i.e., dependence on size, morphology, etc. And, of course, the SEM and EDS analysis could provide reference for potential particle chemical compositions. Therefore, we believe that the SPRM measurement could provide useful information to further separate particles with different hygroscopic properties.

**References**

Xie, Z., Kuai, Y., Liu J., Gui, H., Zhang, J., Dai, H., Xiao, H., Chen, D., Zhang, D., 2020. In Situ Quantitative Observation of Hygroscopic Growth of Single Nanoparticle Aerosol by Surface Plasmon Resonance Microscopy. Anal. Chem. 92(16), 11062-11071. https://doi.org/10.1021/acs.analchem.0c00431.

Kuai, Y., Xie, Z., Chen, J., Gui, H., Xu, L., Kuang, C., Kuang, C., Wang, P., Xu, L., Liu J., Lakowicz, J., Zhang, D., 2020. Real-Time Measurement of the Hygroscopic Growth Dynamics of Single Aerosol Nanoparticles with Bloch Surface Wave Microscopy. ACS Nano. 14(7), 9136-9144. https://doi.org/10.1021/acsnano.0c04513.

---

## Author Comment (AC2)

**Manuscript No.**: ACP-2022-666

**Title**: Atmospheric nanoparticles hygroscopic growth measurement by combined surface plasmon resonance microscope and hygroscopic-tandem differential mobility analyzer

We thank the anonymous referees for their valuable and constructive comments/suggestions on our manuscript. We have revised the manuscript accordingly and please find our point-to-point responses below.

**Comments by Anonymous Referee #2:**

*General Comments:*

*Review to "Atmospheric nanoparticles hygroscopic growth measurement by combined surface plasmon resonance microscope and hygroscopic-tandem differential mobility analyzer". The authors present combined measurements of aerosol hygroscopic growth using an HTDMA and a new SPRM apparatus, targeting at the hygroscopic behavior of bulk aerosols and single particles of 100, 150, and 200 nm, respectively. Combined with the classification of chemical component from SEM-EDX investigations, the authors try to link the single-particle hygroscopicity of different chemical components and the non-uniform distribution of the bulk aerosol hygroscopic growth factor. This method is novel and fits into the scope of ACP. However, the significance of this combined hygroscopic growth study needs to be furtherly clarified, and more detailed information should be provided to make it a solid work. The reviewer recommends accepting this manuscript after addressing the following comments.*

**Response:** We thank the reviewer for the constructive suggestions and comments. Point-to-point responses to comments and questions are detailed below. Following the reviewer's suggestions, we organized the manuscript in the clearer way and clarified the significance of combined SPRM and HTDMA measurements for particle hygroscopic growth studies. The new results and discussions are now included in the revised manuscript.

*Major comments:*

1)  *What is the scientific question the authors want to address, based on the coupled SPRM and HTDMA measurement? To me, it looks like a closure study of aerosol hygroscopic properties based on the single-particle GF quantification and the bulk GF distribution for ambient aerosols. What type of additional knowledge it provides regarding the mixing state of aerosol chemical components?*

**Response:** We thank the reviewer's comment. Yes, it looks like a closure study but investigates aerosol hygroscopic properties from very different perspectives, i.e., single-particle and bulk aerosols. As the hygroscopic properties of ambient aerosols are not uniform but spreads among particles of the same size. To better understand the contribution of different aerosol components, we conduct combined hygroscopic growth measurements using a SPRM-ARI and an HTDMA and establish a link between the apparent hygroscopic properties of single particles and bulk aerosols, thereby providing more information about particle chemical composition and hygroscopic properties. We first identified individual particles with distinct hygroscopic growth behaviors from the SPRM single-particle probing and classified those particles into different categories including non-hygroscopic (NH), less-hygroscopic (LH), and more-hygroscopic (MH). The chemical compositions of individual particles were identified using SEM/ESD analysis, and the results likely agree with the apparent hygroscopic properties. Next, the mean growth factor (GF) of the three categories can be utilized to reproduce the GF distribution obtained from the HTDMA measurement, such that the number fractions of the three categories can be retrieved. We clarified this in the revised manuscript.

2)  *The authors demonstrate the classification of the four groups (i.e., EC, fly ash, OC and AS+OC) in terms of ambient aerosol chemical components, based on the EDS mapping of SEM images. Please clarify the detailed approach of the classification and quantify how representative it is.*

**Response:** In this study, we take advantage of the SEM image and EDS spectra of individual particles, the relative abundance of key elements (e.g., C, O, and S) can be quantified for each particle. According to the particle morphology and elemental composition, the individual particles can be classified into different categories (Kirpes et al., 2018), i.e., organic carbon (OC), soot (mainly elemental carbon), fly ash and secondary aerosols (mainly OC and sulfate). The SEM and EDS analysis provides reference for potential particle chemical compositions.

3)  *The low resolution of Fig. 3 makes the particle imaging at different RH levels blurred. Please provide a clear figure or equivalent statistics supporting the derivation of GF from GI intensity.*

**Response:** According to the reviewer's suggestion, we provided the SPRM-ARI figures (i.e, Fig. 3) with higher contrast, and now the variation of gray intensity under different RH conditions can be clearly observed.